# Symbiotic Husbandry of Chickens and Pigs Does Not Increase Pathogen Transmission Risk

**DOI:** 10.3390/foods11193126

**Published:** 2022-10-08

**Authors:** Emma Kaeder, Samart Dorn-In, Manfred Gareis, Karin Schwaiger

**Affiliations:** 1Chair of Food Safety and Analytics, Faculty of Veterinary Medicine, LMU Munich, Schoenleutnerstr. 8, 85764 Oberschleissheim, Germany; 2Unit of Food Hygiene and Technology, Institute of Food Safety, Food Technology and Veterinary Public Health, University of Veterinary Medicine, Veterinaerplatz 1, 1210 Vienna, Austria

**Keywords:** *Campylobacter* spp., *E. coli*, free-range rearing system, MALDI-TOF MS, FT-IR, animal welfare

## Abstract

A symbiotic or mixed animal husbandry (e.g., pigs and chickens) is considered to have a positive effect for animal welfare and sustainable agriculture. On the other hand, a risk of infection and transmission of microorganisms, especially of zoonotic pathogens, between animal species may potentially occur and thus might increase the risk of foodborne illnesses for consumers. To prove these assumptions, two groups of animals and their environmental (soil) samples were investigated in this study. Animals were kept in a free-range system. In the first group, pigs and chickens were reared together (pasture 1), while the other group contained only pigs (pasture 2). During a one-year study, fecal swab samples of 240 pigs and 120 chickens, as well as 120 ground samples, were investigated for the presence of *Campylobacter* spp., *Salmonella* spp. and *E. coli*. Altogether, 438 *E. coli* and 201 *Campylobacter* spp. strains were isolated and identified by MALDI-TOF MS. *Salmonella* spp. was not isolated from any of the sample types. The prevalences of *Campylobacter coli* and *C. jejuni* in pigs were 26.7% and 3.3% in pasture 1 and 30.0% and 6.7% in pasture 2, while the prevalences of *C. coli* and *C. jejuni* in chickens from pasture 1 were 9.2% and 78.3%, respectively. No correlation between the rearing type (mixed vs. pigs alone) and the prevalence of *Campylobacter* spp. was observed. All swab samples were positive for *E. coli*, while the average prevalences in soil samples were 78.3% and 51.7% in pasture 1 and 2, respectively. Results of similarity analysis of the MALDI-TOF MS spectra (for *C. coli*, *C. jejuni* and *E. coli*) and FT-IR spectra (for *E. coli*) of the same bacterial species showed no recognizable correlations, no matter if strains were isolated from chickens, pig or soil samples or isolated at different sampling periods. The results of the study indicate that the symbiotic husbandry of pigs and chickens neither results in an increased risk of a transmission of *Campylobacter* spp. or *E. coli*, nor in a risk of bacterial alteration, as shown by MALDI-TOF MS and FT-IR spectra. In conclusion, the benefits of keeping pigs and chickens together are not diminished by the possible transmission of pathogens.

## 1. Introduction

In recent years, the meat industry has increasingly gained the interest of society. Partially triggered by scandals led by buzzwords such as zoonotic diseases (e.g., *Salmonella* spp., *Campylobacter* spp. and enterohemorrhagic *E. coli*), consumers are increasingly taking a critical look at primary production and the downstream stages. In addition to product quality and product safety, the social and ethical aspects of animal husbandry are a major concern [1,2,3].

In many ways, animal husbandry offers a high potential for improvement in animal welfare and sustainability, both ecologically and economically [4]. In many countries, a large part of conventional husbandry types is considered as unsustainable in the long run, such as that declared by the Federal Ministry of Food and Agriculture (Germany) [5]. This knowledge and a changed human-animal relationship have led to a critical rethinking [6]. Additionally, there is a broad support among the population demanding that animals are treated with care and respect and that they are given the opportunity to practice species-appropriate behavior [5].

Meat production takes up a large share in the food sector. This discrepancy between the demand for animal welfare and maximum economic value has led to an urgently needed review of animal welfare standards [7,8]. It is important to respond to this change in the society’s perception by creating new opportunities in animal husbandry [9,10]. 

By keeping pigs and chickens together on the pasture, animal welfare-relevant symbiotic effects and the sustainability of animal husbandry systems can be optimally exploited. The benefits of keeping chickens and pigs together could include, for example, giving the chickens better access to earthworms and other food by having the pigs stir up the soil. For their part, the chickens could provide the pigs with protection from ectoparasites. Another benefit to the chickens could be that the pigs offer them protection from birds of prey such as the goshawk. However, at the same time, it raises the question as to whether this kind of animal husbandry leads to an increased exchange of pathogens and thus to a potentiation of the risk of disease transmission. Since *Campylobacter* spp., *Salmonella* spp. and *E. coli* are considered to be important pathogens in both pigs and chickens and are among the most common foodborne zoonoses in Europe [11], they were chosen as model microorganisms for the tracking investigations in this study. 

*Campylobacter* spp. are gram-negative, microaerophilic bacteria. Campylobacteriosis caused by *Campylobacter* (*C.*) *jejuni* and *C. coli* is the most common bacterial diarrheal disease in humans [12]. They are considered as common zoonotic agents, with contaminated food being the main route of transmission, posing a high risk [13,14,15]. Although the two species mentioned above are not obligately host bound, *C. coli* are more frequently detected in pigs and *C. jejuni* in chickens [16,17].

After campylobacteriosis, the second most frequent, notifiable bacterial gastrointestinal disease in humans is salmonellosis [18]. Like *Campylobacter* spp., not all *Salmonella* serovars are obligately bound to the host. Nevertheless, there is a species-specific clustering of some serovars, e.g., *S.* Typhimurium in humans, pigs and chickens, *S.* Enteritidis in humans and chickens, *S.* Infantis and *S*. Gallinarum in chickens [19,20]. There are various possibilities for the transmission of *Salmonella* spp. within livestock. Depending on the serovar, it can be spread via latently infected animals, contaminated feed or other vectors, e.g., rodents, contaminated objects and birds [21,22]. The most common cause of human infection is the consumption of contaminated animal products [23].

The third investigated bacterial species in this study is *Escherichia coli*. They are gram-negative, facultatively pathogenic, flagellated rod-shaped bacteria that are commonly found in human and animal intestines [24,25]. Due to their ability to rapidly absorb and transfer genetic information, *E. coli* are considered as indicator and reservoir germs. Thus, they are particularly of interest for scientific studies dealing with epidemiological questions [26].

The aim of the study was to find out whether animal husbandry types (pigs and chickens vs. pigs alone) have an influence on the risk of shedding, and transmission of *Campylobacter* spp., *Salmonella* spp. and *E. coli*. Additionally, the isolated bacterial strains were investigated using MALDI TOF MS and FT-IR to see if the spectra are converging over time, which could indicate increased exchange between the animal species.

## 2. Materials and Methods

### 2.1. Study Design (Sampling)

#### 2.1.1. Pre Sampling

A pre-sampling was performed to obtain the prevalence of investigated bacteria in animal and soil samples. Before starting the main experiment, rectal swabs were taken once from pigs (*n* = 10) and cloacal swabs were taken once from chickens (*n* = 10). At this point, the animals were each in their parent stocks and had no contact with each other. In addition, soil samples (*n* = 10) were taken once before the animals went out to pasture. The method of sample collection corresponds to the later applied study procedure (see sample collection, Section 2.2).

#### 2.1.2. Forms of Husbandry

The animals were separated into two different groups, living on different pastures. Both pastures were not previously used for any agricultural purpose for the past ten years. For the study, pasture 1 was used for pigs (35) and poultry (about 250) as mixed husbandry and pasture 2 for pigs only (35; comparison group). Each pasture had an area of 2.5 ha. The distance between both pastures was two meters on each side separated by a double fence. Thus, direct contact between animals from both pastures can be ruled out. All investigated animals received feed from the same producer and the same source of water. Figure 1 shows the structure of each pasture. Pigs (3–5 months old) and chickens (4 weeks old) were obtained from the respective breeding stations of the same farm. They were kept in the pastures until reaching age of slaughtering, namely 12 months for pigs and 5 months for chickens. Then, new animals were continually introduced in the two pastures. Altogether, two pig and three chicken groups were introduced to the corresponding pastures. The whole study was localized in Upper Bavaria, Germany.

### 2.2. Sample Collection

#### 2.2.1. Animals

Rectal and cloacal swabs of 240 pigs and 120 chickens (from 12 monthly sampling runs with the exception of May and June 2020 due to the pandemic situation) were investigated between September 2019 and October 2020. For each sampling run, 10 rectal and 10 cloacal swabs were obtained from pigs and chickens from each pasture. 

Two persons performed the swab sampling of animals. Sterile single-use swabs with Amies transport medium (Sarstedt, Germany) were inserted into the recta of pigs and the cloacae of chickens. The swabs were immediately put into the transport medium, individually labeled, packed in three different disposable bags (pasture 1—pig, pasture 1—chickens, pasture 2—pig), placed in a cooling box and transported to the laboratory within three hours. The animals were randomly selected. To assure that none of the animals was sampled twice, the pigs were marked using a marker pen immediately after the sampling was performed. As for chickens, the poultry coops were closed, and each chicken was released after the sampling procedure.

#### 2.2.2. Soil

A total of 60 soil samples per pasture obtained from 12 sampling runs were investigated at the same time as the animal swab samples. The locations of five sampling sites from each pasture are shown in Figure 1. The soil sampling method was adopted from a procedure developed by the State Office for Nature, Environment and Consumer Protection in North Rhine-Westphalia, Germany [27]. The near-surface soil samples with a sampling depth of 2–4 cm were cut with a hole saw (Wolfcraft^®^ GmbH, Kempenich, Germany), recorded with a diameter of 100 mm. For sampling, the metal cylinder was driven into the ground with a plastic hammer. After the excavation, the soil column in the cylinder (approximately 100 g) was transferred to a 200 mL sterile screw-type beaker (Sarstedt, Germany). Between the individual samples, the hole saw was freed from leftover soil with a knife and then disinfected with 70% alcohol. Samples were placed in a cooling box and transported to the laboratory within three hours. 

### 2.3. Sample Preparation

The bacteriological analysis was started within 3 h after sample collection.

#### 2.3.1. Animal Samples

The 20 rectal and 10 cloacal swabs from each sampling run were processed as individual samples under sterile conditions. In a first step, the swabs were streaked directly on a RAPID’*E. coli* agar (Bio-Rad, Feldkirchen, Germany). This agar is recommended for the enumeration of *E. coli* in water and food [28,29]. The protective cap of the swab was then removed using a sterile scissor, while the swab was put into a sterile disposable tube (Greiner Bio-One, Germany) that was previously filled with 5 mL of buffered peptone water. All 30 tubes containing swabs were closed and shaken for 25 min at 250 rounds/min (FL-3005 varioshake, GFL, Lauda, Lauda-Königshofen, Germany) at room temperature. The “peptone water sample suspension” (PSS) was used as the starting material for the subsequent culturing of *Campylobacter* spp. and *Salmonella* spp. 

#### 2.3.2. Soil Samples

Each of the ten screw cups (Sarstedt, Germany) containing soil samples was opened under a sterile laminar flow workbench. Soil was transferred into a sterile flask and weighed to 10 g, then mixed with 90 mL of peptone water by shaking at 250 rounds/min for 25 min at room temperature. This PSS of soil served as the starting material for the subsequent culturing of all target bacteria.

### 2.4. Bacteriological Investigation

#### 2.4.1. Isolation of *Escherichia coli*

*E. coli* were isolated using the RAPID’*E. coli* 2 agar (Bio-Rad, Germany). While animal swabs were directly streaked on the selective agar, approximately 10 µg of the PSS of soil was transferred onto an agar plate and spread out using a sterile inoculation loop. The plates were aerobically incubated at 37 °C for 24 h. After that, one colony from each positive RAPID’*E. coli* 2 agar was subcultured on agar technical (Oxoid, Wesel, Germany) and incubated under the same conditions. The grown colonies proceeded to species identification/confirmation using MALDI-TOF MS (Bruker Daltoniks, Bremen, Germany) and to spectra analysis using FT-IR (Bruker Daltonik GmbH, Bremen, Germany).

#### 2.4.2. Isolation of *Salmonella* spp.

A pre-enrichment procedure was applied in order to revive the potentially sublethally damaged cells of *Salmonella* spp. For this step, 1 mL of the PSS suspension was transferred to 5 mL of buffered peptone water (Thermo Scientific™, Waltham, MA, USA) and aerobically incubated at 37 °C for 16–20 h. From this pre-enrichment, 0.1 mL was dropped in triplicate onto the Modified Semisolid Rappaport-Vassiliadis (MSRV) medium (Oxoid, Germany) and incubated non-inverted at 42 °C for 24 h. Growth of *Salmonella* spp. on MRSV is indicated when a clear opaque halo has formed around the droplet. For further confirmation steps, material from the rim of the opaque halo was subcultured onto Xylose-Lysine-Tergitol 4 (XLT4) agar (Oxoid, Germany) and Brilliant-green Phenol-red Lactose Sucrose (BPLS) agar (Oxoid, Germany). The agar plates were aerobically incubated at 37 °C for 24 h.

#### 2.4.3. Isolation of *Campylobacter* spp.

Enrichment of the thermophilic *Campylobacter* spp. was primarily performed, starting with transferring 1 mL of the PSS into 9 mL of a Preston selective broth (Carl Roth, Karlsruhe, Germany), followed by incubation under microaerobic conditions (5% O_2_, 10% CO_2_, Anaerocult™ C 2.5 l (Merck, Darmstadt, Germany)) at 42 °C for 48 h. The selective enrichment procedure was used in order to enhance the growth of *Campylobacter* spp. and at the same time to reduce or inhibit the growth of the accompanying microorganisms, which may be present in a high number in fecal swab and soil samples. After incubation, the suspension was filtered through a sterile membrane filter with a pore size of 0.65 μm (VWR, Hannover, Germany). Approximately 10 µL of the flow-through suspension was transferred to a Columbia blood agar containing sheep blood (CBA, Oxoid, Germany) with a disposable loop and was streaked using a 3-loop smear technique. The CBA plates were incubated under microaerobic conditions at 42 °C for 48 h. The grown colonies proceeded to species identification using MALDI-TOF MS.

#### 2.4.4. Species Identification by MALDI-TOF MS

The colonies of bacterial cultures were identified to species level using Matrix Assisted Laser Desorption Ionization—Time of Flight Mass Spectrometry (MALDI-TOF MS). Colonies of pure cultures were extracted using the direct transfer method as described in the Bruker Daltonik User’s manual [30]. An appropriate colony mass on the agar plate was taken using a toothpick and smeared on a ground steel BC target plate. Then, 1 µL of a low-molecular organic matrix solution (saturated solution of a cyano-4-hydroxycinnamic acid in 50% acetonitrile) was added. During the drying process at room temperature, a co-crystallization took place in which the analyte was incorporated into the matrix crystals. The MALDI-TOF MS measurements were performed using a Microflex LT (Bruker Daltoniks, Bremen, Germany). The analysis of the generated data was executed with the Software—Biotyper OC incl. Taxonomy (Version 3.1.66, Bruker Daltoniks, Bremen, Germany) and its automated settings.

#### 2.4.5. FT-IR

The Fourier-transform infrared spectroscopy (FT-IR) measurement was applied to *E. coli* strains because of their role as indicator and reservoir bacteria as described in the introduction. Additionally, some studies related to the application of FT-IR have shown that the stability of the bacterial cell mass remains stable up to 24 h after subculturing [31]. For cell masses grown for shorter/longer periods or in other nutrient solutions, the FT-IR spectra sometimes differ considerably. Therefore, reproducible and meaningful information can only be expected from cell masses obtained under standardized conditions [32,33]. To ensure these standardized conditions during FT-IR measurement, all *E. coli* strains were cultured on the same medium, incubated at the same room temperature for exactly 24 h. The restrained growth of *Campylobacter* spp. did not allow this standardized measurement with the sample size, since the incubation time had to be extended if the growth rate was too slow or the colonies were too small.

For each *E. coli* strain, three biological replicates were prepared for FT-IR measurement. The material from each colony was removed from the agar technical after exactly 24 h of incubation using a 1-µL disposable loop. The amount was equivalent to an overloaded inoculation loop. It is important to note that the cell material was only removed from the confluent growth zone. The cell material was transferred to a 1.5-mL reaction tube that was prefilled with 50 μL ethanol (70%) and four inert metal cylinders (Bruker Daltonik GmbH, Bremen, Germany), then mixed by shaking at 250 rounds for 15 s. The 70% ethanol killed the microorganisms, thus stopping their ongoing metabolic activities. To increase the surface tension of the suspension, 50 μL of deionized water was added. Then, 15 μL of each isolate suspension was pipetted onto three spots (technical replicates) of the 96-well microtiter plate (Bruker Daltonik GmbH, Bremen, Germany). The spots on the plate had to be completely dried at 37 °C in an incubator (approximately 30 min) before they were subjected to FT-IR measurement.

Additionally, a quality control for each FT-IR measurement was required. This was carried out by pipetting 12 µL of each Bruker Infrared Test Standards Solutions (IRTS 1 and IRTS 2) on the same microtiter plate. These two standard solutions are part of the Bruker IR Biotyper kit (Bruker, Bremen, Germany). Finally, FT-IR spectroscopy was performed using an IR biotyper spectrometer (Bruker Daltonik GmbH, Bremen, Germany) according to the instructions of the producer [34]. Briefly, each *E. coli* strain was automatically scanned 64 times. Spectra were acquired up to 1500 cm^−1^ with a spectral resolution of 3 cm^−1^ and an aperture of 10 mm. All 64 spectra obtained from a single strain were automatically combined, resulting in a single spectrum. The analysis of the generated data was carried out using Biotyper software (Bruker Daltoniks, Bremen, Germany, version 1.5.0.90) and its automatic settings. The spectral data were automatically converted to dendrograms using the average mean spectra method that was further used for the statistical analysis (Chi-square test).

### 2.5. Analysis for Similarities 

For each bacterial group, *Campylobacter* spp. and *E. coli*, the similarity of their protein spectra obtained by the MALDI-TOF MS, were analyzed using the clustering program BioNumerics (version 7.6, Applied Maths, Sint-Martens-Latem, Belgium). 

Additionally, the similarity of the *E. coli* strains (isolated from animals, *n* = 240) was investigated by FT-IR spectroscopy. This involves comparing each spectrum within a species to all other spectra recorded using the same protocols and methods. The comparison of two spectra provides a spectral distance value. The more two spectra match, the smaller the spectral distance (Bruker Daltonik GmbH, 2017).

### 2.6. Statistical Analysis

#### 2.6.1. Pearson´s Correlation

To evaluate the correlation between pasture types and the occurrence of the investigated bacteria in pigs/in soil samples, a Pearson correlation coefficient (*r*, Microsoft Excel, 2016) was computed. The strength of the correlation for absolute values of *r* is interpreted as follows; *r* = 0–0.19 is regarded as very weak, 0.20–0.39 as weak, 0.40–0.59 as moderate, 0.60–0.79 as strong and 0.80–1.0 as a very strong correlation (Evans, 1996). Additionally, the *p*-value was calculated based on a two-tailed *t*-test analysis in order to evaluate whether the correlation was statistically significant. In Microsoft Excel, the *p*-value was calculated using the formula = T.VERT.2S (t;df). The T.VERT.2S = two-tailed *t*-test, t = *t*-value and df = degree of freedom. The results were interpreted as statistically significant if the *p*-value was less than 0.05.

#### 2.6.2. Chi-Square

The chi-square test (SPSS software, version 26.0) was used to examine the similarity of genotype identification of *E. coli* with FT-IR spectroscopy with respect to two research questions. First, whether the type of husbandry (mixed/symbiotic vs. control pasture) had a significant influence on the formation of the clusters and, second, whether the animal species had a corresponding influence. Pearson’s chi-square test was calculated with calculation of a continuity correction. An asymptotic significance (two-sided), or *p*-value obtained by chi-square test less than 0.05 means that there is a statistically significant relationship between the factors and clusters. In addition, a likelihood-ratio test was performed. To exclude the possibility of inaccuracies in the chi-square due to small sample sizes, the frequencies to be observed were checked using Fisher’s exact test and the linear correlation was also determined.

## 3. Results

The pre-sampling result showed that the prevalence of *Campylobacter* spp. was 10% in pigs, 20% in chickens, and 0% in soil samples. For *E. coli* it was 100% in all animal samples and 30% in soil samples.

In the main experiment, a total of 639 bacterial strains were isolated from 120 cloacal swabs from chickens, 240 rectal swabs from pigs, and 120 soil samples. These included 438 strains of *E. coli* and 201 strains of *Campylobacter* spp.

*Salmonella* spp. could not be isolated in any of the investigated samples.

### 3.1. Detection and Similarity Analysis of Campylobacter spp.

A total of 201 *Campylobacter* strains were isolated from 51.4% of all investigated animals and 12.5% of all soil samples. The prevalences of these bacteria were 87.5% in chickens and 33.3% (30.0% and 36.7% for pasture 1 and 2, respectively) in pigs. Species identification by MALDI-TOF MS revealed that 43.8% and 56.2% were *Campylobacter coli* and *C. jejuni*, respectively.

Figure 2 shows the distribution in detail and the prevalence of *Campylobacter* spp. in each animal group and in soil samples. The highest prevalence of *C. jejuni* was found in chickens (78.3%), while *C. coli* was mostly found in pigs (28.5% in total, and 27.0% and 30.0% of pigs from pasture 1 and 2, respectively). The prevalences of *C. jejuni* in pigs (3.3% and 6.7% for pasture 1 and 2, respectively) and *C. coli* in chickens (9.2%) were relatively low. The distribution of *C. coli* and *C. jejuni* in soil samples from pasture 1 was similar (12.0% and 10.0%, respectively), as well as in soil samples from pasture 2, where the prevalence was remarkably lower (3.0% and 2.0%, respectively) than pasture 1, but not statistically significant (*p* > 0.05).

According to the Pearson correlation coefficient (*r* value), no correlation between husbandry types and detection of *C. coli* (*r* = 0.03, *p* = 0.57) as well as detection of *C. jejuni* in pigs (*r* = 0.08, *p* = 0,24) was found. For soil samples, a weak positive correlation was found between pasture type 1 and the contamination with *C. coli* and *C. jejuni* in soil (*r* = 0.18, *r* = 0.16, respectively). This means it was more likely to detect both *C. coli* and *C. jejuni* in ground samples from pasture type 1 than from pasture type 2. However, the correlation was evaluated as statistically not significant (*p* = 0.05, and *p* = 0.08, respectively).

Results of a similarity analysis of the protein spectra obtained by MALDI-TOF MS using the clustering program Bionumerics show that *Campylobacter* strains were classified into two major subgroups, *C. coli* and *C. jejuni*. The protein spectra of the same *Campylobacter* species were similar, regardless of their origin (chickens, pigs, or soil samples). Figure 3 shows the protein spectra of *C. coli* and *C. jejuni* isolated from chickens and pigs exemplarily. The peaks of the spectra within the same *Campylobacter* spp. (*C. coli*/*C. jejuni*) did not show any differences among isolates obtained from different samples (pigs/chicken/soil) and from different pastures. The differences of the peaks of MALDI-TOF spectra between *C. coli* and *C. jejuni* were indicated with arrows in Figure 2.

### 3.2. Detection and Similarity Analysis of Escherichia coli

As shown in Figure 4, 438 strains of *E. coli* were isolated from all animal swab samples, while in soil samples they were found in a wide range among sampling runs (between 0% and 100%) without recognizable influence of the duration of grazing. The average prevalence of *E. coli* in soil samples obtained from 12 sampling runs was 78.3% and 51.6% in pasture 1 and 2, respectively. The shedding of *E. coli* in ground samples was further analyzed using Pearson’s correlation coefficient. A weak correlation was found between pasture types and the prevalence of *E. coli* (*r* = 0.28) in ground samples and shedding of *E. coli* on pasture 1 was evaluated as statistically significantly higher than on pasture 2 (*p* = 0.002).

Results obtained from similarity analysis (Bionumerics, Applied Maths) showed that the protein spectra of *E. coli* obtained by MALDI-TOF MS from all sample types have a high similarity (data not shown). The spectra were distributed randomly and were not grouped in sample types (pig/chicken swabs or soil samples) or husbandry types (pigs with chickens vs. pigs alone), but were rather grouped in sampling time (from September 2019 to October 2020). By comparing the spectra obtained from the same sampling run, it was observed that at the beginning of the study (sampling runs one to three) that there was a high diversity in the spectra of *E. coli,* resulting in a high number of clusters. Each cluster included isolates from both husbandry types and/or animal species. In the course of time (sampling runs 4–12), the number of clusters was reduced to one to three, since the spectra of the isolates became more similar, independent of whether they were isolated from chickens or pigs from pasture 1 or pasture 2. According to this analysis, a manifest transformation of a single *E. coli* isolate was not detected. 

In addition, FT-IR spectroscopy was used to analyze whether the spectra of *E. coli* (isolated from animals, *n* = 240) converge over time or whether species-dependent differences persist. *E. coli* cultures that were used for FT-IR spectrometry always showed very uniform and brisk growth within the same cultivation period. Differences between the FT-IR spectra due to technical errors could be excluded by the three biological and three technical replicates or, if necessary, deviating spectra could be sorted out. The comparison of the three technical replicates and the three biological replicates showed that the spectra of one and the same biomass matched. After that, the dendrograms used for statistical analysis were generated as follows: for each sample run, one dendrogram contained the spectra of *E. coli* from the pigs kept in both husbandry types (pasture 1 and 2) and another dendrogram contained the spectra of *E. coli* from the chickens and pigs kept in pasture 1 (mixed husbandry). 

Regarding the interpretation of the created dendrograms, the most important aspect was to find a reasonable cut-off value for distance to see which spectra belong to the same cluster. Since the cut-off value for differentiation at the strain level for bacteria varies slightly in each run, a stable cut-off value of 0.300 was set for differentiation. The cut-off value was set to be as low as possible to achieve a high discriminatory power, but also high enough for the technical replicates to not spread across multiple clusters. As a result, at least one major cluster occurred in all sampling runs, as shown in Figure 5. 

The aim of the cluster evaluation was to find out whether the spectrum of the respective individual animal could be sorted into the corresponding cluster of its group. For this purpose, the largest cluster was determined and it was checked whether predominantly pig or chicken samples occurred in this cluster, and it thus was named the “pig cluster” or “chicken cluster”. Subsequently, the number 1 or 0 was assigned for each individual animal sample. Number 1 meant that the animal sample could be sorted according to its cluster, while 0 meant that the animals were outside the assigned cluster.

The first statistical analysis aimed to find out whether the type of husbandry (mixed/symbiotic vs. control pasture) had a significant effect on the formation of the clusters of *E. coli*. The statistical results revealed that no relationship between factors and clusters could be detected either within each sampling run or when comparing all 12 runs together (Pearson’s chi-square test: asymptotic significance (two-sided) or *p*-value = 0.984, see Table 1). This means that the husbandry type had no influence on the cluster formation of *E. coli*.

The relationship between animal types (chicken/pig) and the formation of clusters was also statistically evaluated. The statistical result in Table 2 shows that no significant effect across all study time points was found (Pearson’s chi-square test: asymptotic significance (two-sided) = 0.283). This indicates that the type of animal (chicken/pig) did not have any influence on the cluster formation of *E. coli* isolates.

All isolates that did not pass the quality check during the FT-IR measurement were automatically sorted out so that the numbers of valid cases used for both statistical analyses were *n* = 225 (Table 1) and *n* = 231 (Table 2).

Furthermore, a multifactorial approach with the generalized linear model (GLM; distribution form of the dependent variable binomial) was applied to investigate the influence of animal species and husbandry type on the distribution of spectra. With the respective results, no statistically significant effects were found (animal species: *p* = 0.256, husbandry: *p* = 0.899).

## 4. Discussion

Topics related to animal welfare of livestock are increasingly discussed in society and have a high influence on consumer decisions regarding whether to buy meat and meat products. A symbiotic or mixed rearing system, in which, for example, two animal species are kept together in the same free ranging area, can significantly contribute to an increased animal welfare status [35]. Another major issue in the critical examination of agriculture is sustainability. Due to global issues such as the ever-growing global population, climate change and an increasing demand for animal protein, the need for more sustainable animal agriculture is more urgent than ever. The pressure to maximize the production of milk and meat has disturbed the equilibrium between feeding and yield, animal welfare, environmental impact and public acceptance [36,37]. More and more ways are being sought to make agriculture more sustainable in the long run and therefore more viable for the future [38]. If the food supply for the growing world population is to be secured in the long term, production systems and consumption patterns will have to change. The challenge is to increase yields on existing lands without leaching it out and losing its fertility [39]. Shared animal husbandry is an approach which is based on the same fundamental idea. By keeping two different species of animals together, only one pasture is needed instead of the usual two, thus increasing the capacity utilization of the space with positive effects on both sustainability and animal welfare. In addition, as observed as a side finding of this study, chickens always spread throughout the pasture and used all of the space for scratching and foraging. This may be a result of their positive feeling of being protected by the pigs from any of their foes such as birds of prey. On the contrary, many different studies have shown that even with a large free-range area, chickens stay very close to their coop out of fear [40,41], and only use the free-range area if they can find protection in the form of a shelter [42]. The findings of the present study clearly demonstrate the protective function of pigs in a mixed husbandry system.

However, the assumption that natural bacterial infection and disease transmission between animal species can increase when different animal species are kept together might impede the implementation of this rearing system for example due to veterinary authority reservations. Therefore, this study was conducted to prove whether the rearing system (pasture 1: chickens and pigs together; pasture 2: only pigs) has an influence on the prevalence of important zoonotic pathogens like *Campylobacter* spp., *Salmonella* spp. and *E. coli*, and whether there is an increased exchange of these isolates, as determined by MALDI-TOF MS and FT-IR spectra. For this purpose, a total of 240 pigs and 120 chickens were investigated between September 2019 and October 2020. Altogether, 438 *E. coli* and 201 *Campylobacter* strains were isolated and identified by MALDI-TOF MS. 

In this study, *Salmonella* spp. could not be isolated in any of the investigated samples.With 8743 cases reported in 2019, salmonellosis is the second most common notifiable bacterial gastrointestinal disease in humans in Europe [11]. Farm animals (e.g., poultry, pigs and cattle) are considered to be the main reservoir, since almost all infected animals do not show any clinical symptoms [23]. A study conducted by the Federal Office of Consumer Protection and Food Safety in Germany (2020) showed that the prevalence of *Salmonella* spp. in caecal content samples of broiler was 2.6% and of broiler turkeys 2.4%, while 4.6% of fecal samples of wild boars and 4.0% of slaughtered fattening pigs were positive for this genus [43]. Although the prevalence of *Salmonella* spp. in farm animals in Germany is relatively low, they were included in the analysis for this study. Within livestock, there are several ways for *Salmonella* transmission, e.g., via latently infected animals, contaminated feed, or other vectors such as rodents, insects, wild birds and contaminated objects [21,22]. Free-range animals, such as in this study, could have a high risk of exposure to these vectors. Additionally, various studies have shown that free-range chickens have a higher prevalence of *Salmonella* spp. [44,45]. On the other hand, once *Salmonella* spp. entered the crops, the transmission rate was much lower in free range and especially in organic farming systems since there is more space available for each animal [46], and probably due to the better welfare aspects that could lead to a higher immune status of animal herds [47]. 

Thermophilic *Campylobacter* spp. could be detected in both pigs and chickens with a relatively similar prevalence to a study carried out in Bavaria (Germany) [48]. In this study, the detection rate of *Campylobacter* spp. in pigs (33,3% in total, 30% in pasture 1 and 36,7% in pasture 2) is slightly lower than in the above-mentioned study (36 %) and is considerably lower than the prevalence detected in other regions such as the Netherlands (46% [16] and 85% [49]). In a study from the United Kingdom, the prevalence of *Campylobacter* spp. is variable depending on the health status of animals, e.g., 77% for sick pigs compared to 44% for healthy pigs [3]. However, it should be noted that apart from ours and the Bavarian prevalence study, all the above-described studies collected the samples at the postmortem stage at the slaughterhouse. Stress and conditions during transport of animals to the slaughterhouse can increase the susceptibility of animals to the disease as well as the risk of disease transmission, possibly explaining the high prevalence of *Campylobacter* spp. in slaughtered pigs, as found in the mentioned studies. In addition to the moderate prevalence of *Campylobacter* spp. in pigs, a high colonization with thermophilic *Campylobacter* spp. (88%) in the chicken group was observed and is similar to data previously collected in Bavaria (75%, [48]). Regarding the bacterial species, *C. jejuni* and *C. coli* show a very different prevalence in the respective animal species in this study. The high prevalence of *C. jejuni* in poultry (over 78%) is consistent with previous reports, considering it as the most commonly detected *Campylobacter* species in chickens and as a natural gut inhabitant [16]. The low detection rate (5%) of *C. jejuni* and the predominance of *C. coli* in pigs are also consistent with the results of numerous studies [17,50,51]. 

The correlation of husbandry types (pasture 1 vs. pasture 2) and the risk of infection with *Campylobacter* spp. was analyzed. Pigs that were in close contact with chickens (pasture 1) have a risk of infection with *C. coli* similarly high to pigs that were kept alone (control group, pasture 2). However, pigs kept in pasture 2 showed a weak correlation to the risk of infection with *C. jejuni*, which is the species that is more frequently found in chickens. The prevalence of *C. jejuni* in the present study was higher in the pigs kept alone than in the pigs kept together with chickens (7% vs. 3%, respectively). Similar results were observed in Denmark, where pig herds kept alone or together with cattle have a tendency of increasing infection with *C. jejuni* than pig herds kept with poultry (i.e., 7.8%, 12.8%, and 4.4% of investigated pig herds, respectively) [50]. In this context, it may be possible that *C. jejuni* has adapted itself to invade other animal species when its specific host (poultry) is not present.

The shedding of *Campylobacter* spp. into soil/ground of pastures was additionally investigated. The prevalences of both *Campylobacter* species in soil samples from pasture 1 were higher than in soil samples from pasture 2. This may be due to the higher concentration of animals in the pasture (35 pigs and 250 chickens in 5 ha for pasture 1, and only 35 pigs for pasture 2). However, the difference was evaluated as statistically non-significant. According to the results of this study, it can be concluded that being kept on pasture 1 (pigs and chickens on mixed husbandry) did not increase the risk of infection of pigs with *Campylobacter* spp. compared to being kept on pasture 2 (pigs kept alone). 

The cluster analysis of protein spectra of *Campylobacter* strains (*n* = 201) obtained by MALDI-TOF MS show that the strains were not sorted into groups based on husbandry, but solely into two groups according to the species *C. jejuni* and *C. coli*. The single spectra of the same *Campylobacter* species (*C. coli*/*C. jejuni*) show no differences between those of the pigs/chickens from pasture 1 (mixed husbandry) to the spectra of the pigs from pasture 2 (control group). Since there was no contact between the chickens (pasture 1) and the pigs of the control group during the project, transmission by direct contact can be ruled out. This result confirmed that no alteration regarding the protein composition of a single *Campylobacter* spp. was detected using this method, which does not indicate an increased exchange of these pathogens.

*E. coli* are mostly considered as harmless commensals, but this species also includes pathogenic variants that are associated with a variety of infections in humans and animals. They can be classified into non-pathogenic, commensal, intestinal pathogenic and extraintestinal pathogenic strains. *E. coli* exhibit a very flexible genome that quickly acquires genetic information horizontally. The genomic region contributes to the rapid evolution of variants [52]. Because of this resulting wide range of phenotypes, *E. coli* is a well-suited model organism for tracking studies. Pronounced genomic plasticity leads to a large variability. Other genomic changes such as DNA rearrangements and point mutations can also constantly alter the genome content and thus the fitness and competitiveness of individual variants in specific niches [53,54]. *E. coli* were isolated from all animal samples (*n* = 360). The shedding of *E. coli* in ground samples of pasture 1 (78.3%) was statistically significantly higher than of pasture 2 (51.6%), which may be the result of the higher concentrations of animals in pasture 1, as described in the discussion part for *Campylobacter* spp. By using protein spectrum analysis, the change of an individual strain and the formation of strain clusters can be recognized; thus, their spectra obtained by MALDI-TOF MS and from FT-IR proceeded to similarity analysis and the data was statistically evaluated. The mass spectrometry analysis was applied in this study, since previous studies have shown it to be highly reliable in terms of discriminatory power and the identification accuracy of microorganisms [33,55,56,57]. Additionally, it requires less material and cost and is rather easy to be conducted with a high number of samples. It may be noted that the results obtained could be extended in subsequent studies using next generation sequencing (NGS) or whole genome sequencing. One possibility would also be the combined and complementary NGS and MALDI-TOF MS techniques for bacterial characterization [58]. However, it was already mentioned in some studies that the 16S rRNA gene, which was often used for the NGS analysis, is rather insufficient at differentiating bacteria down to species level [59]. Thus, using this gene, the differentiation between *C. jejuni* and *C. coli* and between *E. coli* strains might also not be possible [60]. Therefore, specific gene sequences have to be properly selected for the genome analysis.

MALDI-TOF MS spectra of *E. coli* strains isolated within the same sampling run showed a high similarity. Subsequently, the spectra of all *E. coli* isolates (*n* = 438) were clustered according to the sampling time. Similar results were obtained by FT-IR analysis, indicating that the husbandry types (symbiotic living of chickens and pigs vs. pigs alone) and animal species (pigs vs. chickens) did not have any influence on the cluster formation of FT-IR spectra of *E. coli* isolates. Since an alteration of *E. coli* strains isolated from both animal species and husbandry types was not detected, an increased risk for pathogen exchange due to the symbiotic animal husbandry could not be observed in the one-year study period. However, it has to be mentioned that a methodological limitation of the study relates to the number of investigated colonies per plate. As described in the section material and methods, only one colony of *Campylobacter* spp./*E. coli* per culture plate was investigated by MALDI-TOF MS and FT-IR. In a single animal, there could be different bacterial strains. In this context, the observed effect might have been more pronounced if more colonies had been sampled.

Altogether, traditional culturing and state-of-the-art-methods (MALDI-TOF MS, FT-IR and similarity analysis) were applied to evaluate whether there was a risk of increasing disease transmission between two animal species that were kept together for one year. The results indicate that there is no species barrier regarding the transmission of *Campylobacter* spp. and *E. coli* between pigs and chickens. The prevalences of both *Campylobacter* spp. in both animal species are similar to the results of other studies conducted in the same region (Bavaria, Germany). Additionally, a high prevalence of *C. jejuni* in chickens did not result in a high infection rate of this bacteria in pigs raised in the same pasture. Furthermore, the characteristic alteration of *E. coli* was neither observed in the strains originally isolated from pigs or from chickens.

In terms of food safety, it can be concluded that keeping these animals together in free-ranging husbandry does not increase disease susceptibility and transmission regarding *Campylobacter* spp. and *E. coli*. Subsequently, meat and their products from mixed animal husbandry have no additional risk of being contaminated with pathogens (*Campylobacter* spp., *Salmonella* spp.) and indicator bacteria (*E. coli*). The most important factors when aiming to keep infection rates at a low level are the hygienic management of the animal herd, farm biosecurity, and the density of animals. This study was conducted under optimal conditions, where the animals had plenty of space (the legal requirements for access of chickens to open-air runs (broilers) are 4 m^2^ (organic) or 2 m^2^ (conventional) [61]), and were raised on pastures that have not been used for a long time. To verify the results obtained in this study, further investigations are required, for example, under the condition that stocking density is increased and/or when the pastures have been continually used for rearing animals.

## 5. Conclusions

This study was conducted to investigate the influence of symbiotic animal husbandry on the risk of bacterial transmission between pigs and chickens and the risk of the exchange of bacterial isolates between both animal species. The results do not indicate an increased risk of transmission for pigs when they are kept together with chickens in a mixed husbandry system (pasture 1) compared to a pasture with pigs alone (pasture 2). The prevalence of *Campylobacter* spp. in pigs was 30.0% in pasture 1 and 36.7% in pasture 2, and 0% regarding *Salmonella* spp. and 100% for *E. coli* for both pastures. Results obtained by similarity analysis of the MALDI-TOF MS and FT-IR spectra show that husbandry types and animal species did not have any influence on the cluster formation of *Campylobacter* spp. and *E. coli* strains, indicating that protein alteration of isolates of both bacterial species did not occur to a significant extent during the studied period. Therefore, in addition to the highly positive effects on animal welfare and sustainability associated with the symbiotic rearing system, a higher risk of transmission of the investigated pathogens was not ascertained. Neither the composition of the animal groups nor the duration of grazing rearing had a significant influence on the similarity or exchange of individual pathogens in this study. Thus, the advantages of keeping pigs and chickens together under good grazing conditions are not diminished by the possible transmission of pathogens.

## Figures and Tables

**Figure 1 foods-11-03126-f001:**
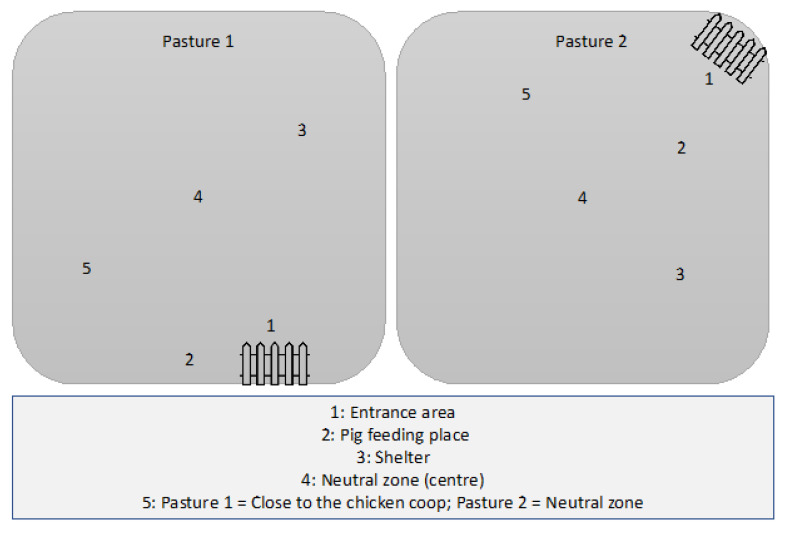
Schematic layout of the pastures and the soil sampling (1 to 5). Pasture 1: pigs and chickens; pasture 2: pigs alone.

**Figure 2 foods-11-03126-f002:**
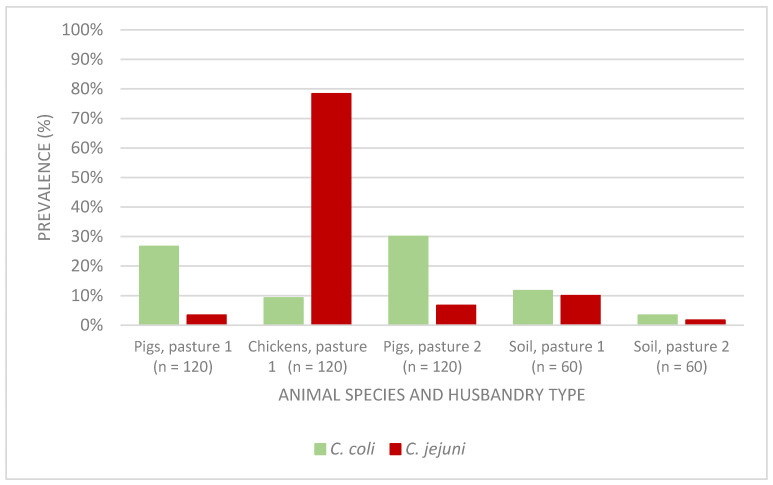
Prevalence of *Campylobacter* spp. in animal and soil samples from two husbandry types. Pasture 1: pigs and chickens were kept together (mixed husbandry). Pasture 2: pigs alone.

**Figure 3 foods-11-03126-f003:**
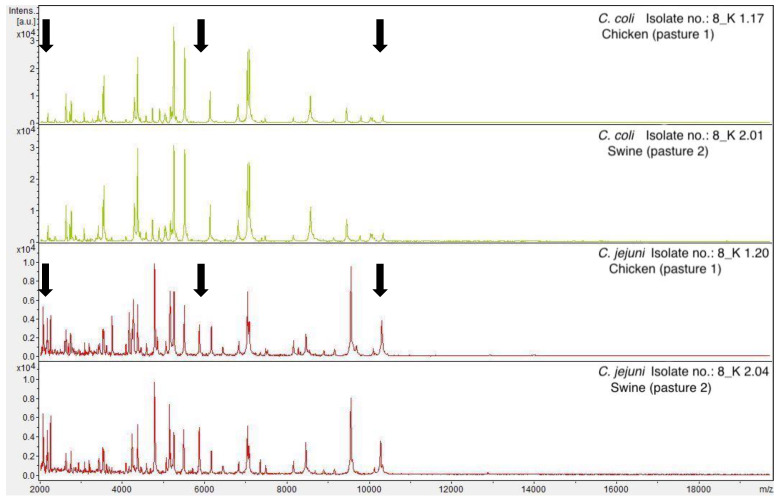
Examples of MALDI-TOF MS mass spectra of *C. coli* and *C. jejuni* isolated from chickens and pigs. Arrows indicate peaks that are absent or present in both species.

**Figure 4 foods-11-03126-f004:**
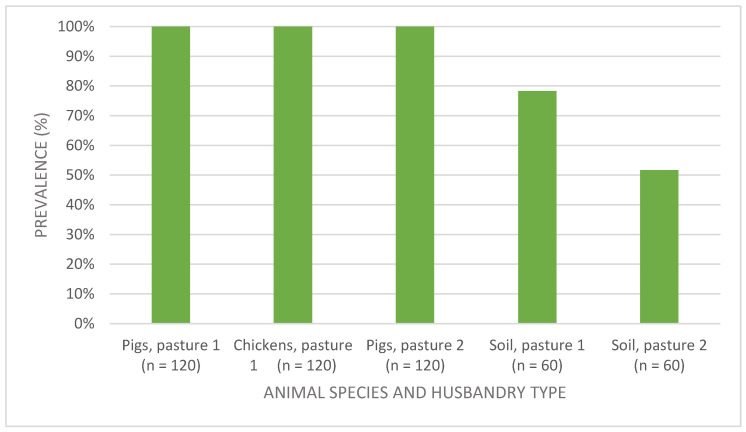
Prevalence of *Escherichia coli* in animal and soil samples from two husbandry types. Pasture 1: pigs and chickens were kept together (mixed husbandry). Pasture 2: pigs alone.

**Figure 5 foods-11-03126-f005:**
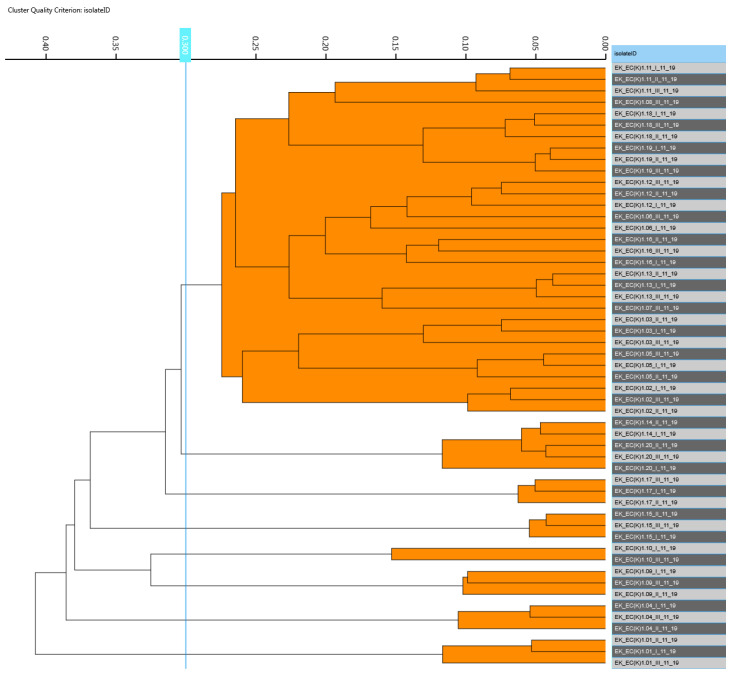
Example of a dendrogram of FT-IR spectra obtained from *E. coli* strains isolated from pigs and chickens kept in pasture 1 (mixed husbandry; third sampling run). The blue line indicates the stable cut-off value of 0.300. A main cluster is in the upper horizontal plane. The right side, highlighted in dark and light gray refers to *E. coli* strains coming from animals. The abbreviation, for example “EK_EC(K)_1.11_I_11_19” stands for: EK = name of author; EC(K) = *E. coli* (K = cloacal) 1.11 = Pasture 1, chicken no. 1 (no. 01–10 = pigs, no. 11–20 = chickens); I = first biological replicate; 11_19 = month November and year 2019.

**Table 1 foods-11-03126-t001:** Chi-square test (FT-IR dendrograms). Influence of husbandry type on the cluster formation of *E. coli* isolated from pigs from pasture 1 (*n* = 120) and pasture 2 (*n* = 120).

Total	Value	Degree of Freedom	Asymptomatic Significance z (Two-Sided)	Exact Significance z (Two-Sided)	Exact Significance z (One-Sided)
Pearson’s chi-square test	0.000	1	0.984		
Continuity correction	0.000	1	1.000		
Likelihood-ratio test	0.000	1	0.984		
Fisher’s exact test				1.000	0.551
Linear correlation	0.000	1	0.984		
Number of valid cases	225				

**Table 2 foods-11-03126-t002:** Chi-square test (FT-IR dendrograms): Influence of animal species on the cluster formation of *E. coli* isolated from pigs (*n* = 120) and chickens (*n* = 120) from pasture 1.

Total	Value	Degree of Freedom	Asymptomatic Significance z (Two-Sided)	Exact Significance z (Two-Sided)	Exact Significance z (One-Sided)
Pearson’s chi-square test	1.153	1	0.283		
Continuity correction	0.868	1	0.351		
Likelihood-ratio test	1.154	1	0.283		
Fisher’s exact test				0.321	0.176
Linear correlation	1.148	1	0.284		
Number of valid cases	231				

## Data Availability

Data is contained within the article.

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
