# Peer review of "Symbiotic Husbandry of Chickens and Pigs Does Not Increase Pathogen Transmission Risk"

_foods, 2022, doi:10.3390/foods11193126_

Round 1
Reviewer 1 Report
Comments and Suggestions for Authors
The aim of the study was to find out whether animal husbandry types have an influence on the risk of shedding, and transmission of Campylobacter spp., Salmonella spp. and E. coli.
The work is relevant and has potential for practical application, but some clarifications and additional information are needed.
1- Before placing animal n the experiment, was any study done to verify the prevalence of contamination?
2- Was it a single collection per animal? Is this representative?
3- Was there control of possible contamination from the feed that the animals received?
4 - Besides the soil, is there the possibility of another source of contamination? And was the water used analyzed?
5 - Could the continued introduction of animals have interfered with the results?
6 - What was the diluent solution used to place the swab?
7 - There is a problem at line 128, 292, 315, 316, 325, 368, 385, 388, 394; please check.
8 - Check the entire text and correct scientific names that are not in italics.
9 - Center table 1 and 2.
10 - Improve the discussion by addressing more the practical application of the study as well as its limitations.
Reviewer 2 Report
Comments and Suggestions for Authors
The study is well written and laid out and I applaud the authors for that.
As the researchers rightfully claimed in the discussion, C. jejuni is more prevalent in poultry, and C. coli is more prevalent in pigs. Except for a few studies, this is the norm. However, both also carry the other specie in a less prevalent manner and this is also a common feature; C. coli for poultry and C. jejuni for pigs. Within the results and the discussion the researchers did not find significant differences and thus came to the conclusion that with Campylobacter and E. coli no significant differences were found, with Salmonella excluded as no strains were found.
It is expected that the pasture with more animals will have an increased E. coli and Campylobacter load in the soil as there was more animal faecal output, but then the researchers could not find significant differences in the prevalence of the species between the pasture animals. Having just collected one colony per culture and E. coli being a commensal bacteria, I am not convinced this conclusion carries any weight. An organism that travelled from one host to another may have had genetic differences and since they co-exist with other strains, may have been missed.
Basically, I am not sure if the study design is sufficiently accurate to determine the objectives of this study.
References
1. There is a general tendency in this manuscript of using very old references on themes that are well researched. I would urge the researchers to update these so that the work fits better in the current global knowledge base.
2. Reference 19 is incomplete
Method
11. Please provide an ethical statement of approved animal trials and a reference number.
22. A major limitation is that only one colony was collected per culture plate. Both E. coli and Campylobacter can have multiple clonal types on a plate and thus when wanting to search for transmission it’s best to collect a minimum of five colonies. The researchers may have missed some strains.
3. The cloacae of chickens may not be the optimal source for intestinal bacteria, I would suggest that fresh faecal droppings for the next study may yield higher results.
44. Please clarify if the MALDI-ToF ranges were always accurate on both genus and specie levels and what percentages. Having used similar techniques some years ago, the reviewer found that the Bruker database was often restricted for Campylobacter having weak outputs.
55. Only the FT-IR analyses were able to detect some similarity output although I have not seen this used as a gold standard for cluster determination and I wondered if the authors can clarify its level of accuracy for this function.
6. Please make sure all controls are present for the tests conducted.
Results
1 1. Line 302- 305: While the study aims to find differences in husbandry housing of two species compared to one specie, C. jejuni was significantly more prevalent in pigs kept alone than in pigs kept with poultry that generally have a higher C. jejuni prevalence.
Discussion
1. Line 443: The absence of Salmonella should already be mentioned in the Result section as it is first mentioned in the Discussion section.
Figures
Figure 3 is perhaps redundant unless it is better described. MALDI-ToF has been around for some time, so we can consider excluding it.
Round 2
Reviewer 2 Report
Comments and Suggestions for Authors
Dear Authors,
The reviewer appreciates some of the changes that were made. I would like to highlight some important outstanding issues:
1. As mentioned in the first review an ethical statement has to be produced as animals are involved and are according to international guidelines. If none was needed it has to be explicitly stated that way, but in many countries, the collection of any animal-derived products needs ethics clearance (including faeces), and in this case, the animals are actually handled and are part of an experimental trial. Please provide details on this issue.
2. Line 317-318: The following statement: "C. jejuni was significantly more prevalent in pigs kept alone than in pigs kept with poultry that generally have a higher C. jejuni prevalence" needs clarity. What could have contributed to this significant difference? Perhaps some mention is needed in the Discussion as the focus was placed on the mixed pasture and not on what happened with this group.
3. Please clarify Figure 5: Are the authors indicating different clusters of E. coli according to the source, the information is not visible. Suggest that authors provide more description in capture.
4. Is another limitation to this study not the less accuracy of cluster determination of mass spectrometry technology? Would other technologies (molecular) not provide a more accurate conclusion?
5. Would still suggest that some of the very old references be updated where relevant.
